# 3D Geophysical Predictive Modeling by Spectral Feature Subset Selection in Mineral Exploration

**Bahman Abbassi** [1,*] **, Li-Zhen Cheng** [1] **, Michel Jébrak** [2] **and Daniel Lemire** [3]

[1] Institut de Recherche en Mines et en Environnement (IRME), Université du Québec en Abitibi-Témiscamingue (UQAT), Rouyn-Noranda, QC J9X 5E4, Canada
[2] Département de la Science de la Terre et de l'Atmosphère, Université du Québec à Montréal (UQAM), Montreal, QC H2X 1L4, Canada
[3] Département de Science et Technologie, Université TÉLUQ, Montreal, QC G1K 9H5, Canada
[*] Correspondence: bahman.abbassi@uqat.ca

**Abstract:** Several technical challenges are related to data collection, inverse modeling, model fusion, and integrated interpretations in the exploration of geophysics. A fundamental problem in integrated geophysical interpretation is the proper geological understanding of multiple inverted physical property images. Tackling this problem requires high-dimensional techniques for extracting geological information from modeled physical property images. In this study, we developed a 3D statistical tool to extract geological features from inverted physical property models based on a synergy between independent component analysis and continuous wavelet transform. An automated interpretation of multiple 3D geophysical images is also presented through a hybrid spectral feature subset selection (SFSS) algorithm based on a generalized supervised neural network algorithm to rebuild limited geological targets from 3D geophysical images. Our self-proposed algorithm is tested on an Au/Ag epithermal system in British Columbia (Canada), where layered volcano-sedimentary sequences, particularly felsic volcanic rocks, are associated with mineralization. Geophysical images of the epithermal system were obtained from 3D cooperative inversion of aeromagnetic, direct current resistivity, and induced polarization data sets. The recovered cooperative susceptibilities allowed locating a magnetite destructive zone associated with porphyritic intrusions and felsic volcanoes (Au host rocks). The practical implementation of the SFSS algorithm in the study area shows that the proposed spectral learning scheme can efficiently learn the lithotypes and Au grade patterns and makes predictions based on 3D physical property inputs. The SFSS also minimizes the number of extracted spectral features and tries to pick the best representative features for each target learning case. This approach allows interpreters to understand the relevant and irrelevant spectral features in addition to the 3D predictive models. Compared to conventional 3D interpolation methods, the 3D lithology and Au grade models recovered with SFSS add predictive value to the geological understanding of the deposit in places without access to prior geological and borehole information.

**Keywords:** independent component analysis; 3D modeling; spectral feature subset selection

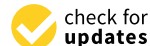



## 1. Introduction

A fundamental problem in integrated geophysical interpretation is the proper geological understanding of multiple inverted physical property images, which demands high dimensional techniques for extracting geological information from modeled physical property images. The study of seismic attributes is an example of high-dimensional pattern recognition that seeks to extract relevant geological features from broadband seismic data [1–7]. Non-seismic data interpretations in mineral exploration involve a similar classification problem, where extracting 3D geological information from inverted potential field data is a computational challenge.

Several techniques are available for feature extraction in the high dimensional space, and this computational challenge can be seen as a dimensionality increase problem that

consequently leads to an overload of information [8–10]. For example, the spectral decomposition of geophysical images provides a robust way for feature extraction in the frequency domain [11–13]. However, the underlying patterns inside the high-dimensional images are only related to certain frequencies, and most decomposed spectra are redundant. The question is which frequency or frequency ranges are geologically relevant. This computational challenge can be seen as a dimensionality reduction problem, in which we try to extract the best representative components of high-dimensional images to facilitate visual interpretations and machine learning optimization [14,15].

Conventionally, high redundancy in the spectral domain is treated by incorporating principal component analysis (PCA) to extract the principal components of multiple images [14,16]. PCA is an unsupervised algorithm that linearly transforms the multivariate datasets into new features called Principal Components (PCs) by maximizing the variance of the input data [14,16]. However, the uncorrelatedness of principal components is a weaker form of independence; therefore, PCA is not a good choice for separating overlapped features inside multiple physical property images [17]. Alternatively, independent component analysis (ICA), precisely the negentropy maximization approach, has been proposed as a robust tool for separating background lithotypes from alteration events [17,18]. Unlike PCA, ICA performs beyond the Gaussian assumption and incorporates higher-order statistics to find maximally independent latent features [14,16].

As expected, the spectra of the ICs still contain certain hidden overlapped features in different frequencies. A similar problem has been addressed by Honório et al. [19] to visualize seismic broadband data in different frequencies. They used ICA to maximize the non-gaussianity of the decomposed seismic frequency volume and stacked the resulting ICs in three red, green, and blue (RGB) channels. The unified RGB image delineated a hidden sedimentary channel system [19]. However, this approach lacks automation and involves manually selecting and stacking representative ICs of the decomposed seismic data. The extraction of spectral features provides more detailed input features for the machine learning algorithm; however, the existence of redundant features decreases the performance of the learning process. Finding a way to detect these redundancies and avoid them can improve machine learning predictions. Moreover, it is challenging to decide how much dimensionality reduction is necessary to represent the most prominent spectral features to machine learning algorithms.

Feature subset selection (FSS) provides a powerful tool for adaptive dimensionality reduction and feature learning [15,20–22]. SFSS reduces the size of a large set of input features to a new set of relatively small and representative input features that improve the accuracy of the artificial neural network (ANN) prediction, either by decreasing the learning speed and model complexity or by increasing generalization capacity and classification accuracy [23]. Therefore, unlike conventional statistical dimensionality reduction methods that are blind source separation algorithms, there is a criterion for the selection of the best features that can be expressed as an adaptation of the learning process in the light of the optimization of both neural synaptic weights and the number of selected input features [21–23].

This study's spectral feature subset selection (SFSS) procedure is seen as a multi-objective optimization problem in which both the ANN weights and the number of the selected features are updated in different iterations. Several heuristic algorithms are known to solve these multi-objective optimization problems [24–26]. We use a bi-objective genetic algorithm (GA) optimization to regularize the ANN weights and find the most adaptable combination of input spectral features in the supervised machine learning procedure. The advantage of this approach to the conventional representative learning algorithms is that it can direct the learning of ANN in a way that only a relevant set of input features are allowed to be used during the learning process.

The main objective of this study is to present a robust algorithm to train ANN with automatically selected spectral features to predict geological targets such as mafic to felsic 3D lithological variations and 3D Au-grade distribution. The predicted outputs of the

SFSS procedure are the estimated geological targets and selected sets of spectral features related to each geological target. The study also aims to show that the selected features and predicted targets also provide a unique way for interpreters to see which spatial and spectral features are prominent in relation to the geological targets.

## 2. Materials and Methods

We developed an algorithm that automatically selects the best representative components based on multi-objective machine learning optimization. Our self-proposed 3D SFSS algorithm works for dimensionality reduction and separating high-dimensional spectral overlapped features. However, SFSS implementation in this study demands several pre-processing stages, including 3D inversion of geophysical data and 3D feature extraction methods. The accuracy of inverted physical property images is critically important in the subsequent feature extraction and selection procedures. 3D inversion algorithms are dedicated to tackling this problem [27–30]. Even if the geophysical inversion provides the most accurate physical property distributions, each physical property image provides only partial information about subsurface geology. This phenomenon can be seen as a linear mixing process, and ICA can help to reconstruct the hidden geological features from multiple physical property images [17,18]. The advantage of this method is that it can separate the host geology from alteration overprints in multidimensional physical property space [17].

Subsequently, spectral decomposition reveals many latent features that are not properly visible in the spatial domain. However, the statistical interdependence of the decomposed images is a major difficulty in extracting and selecting geological information. The large volumes of decomposed spectra also limit the spectral decomposition's efficiency and demand effective dimensionality reduction methods to prevent information overload. Our self-proposed SFSS algorithm solves these high-dimensional problems in five stages (Figure 1):

1. 3D inversion of geophysical data.
2. Separation of physical properties through ICA (spatial feature extraction).
3. Spectral decomposition of ICs through a continuous wavelet transform (CWT).
4. Dimensionality reduction and separation of raw spectral features through ICA (spectral feature extraction).
5. SFSS based on a supervised feature selection algorithm optimized by a multi-objective GA optimization (spatiospectral feature selection).

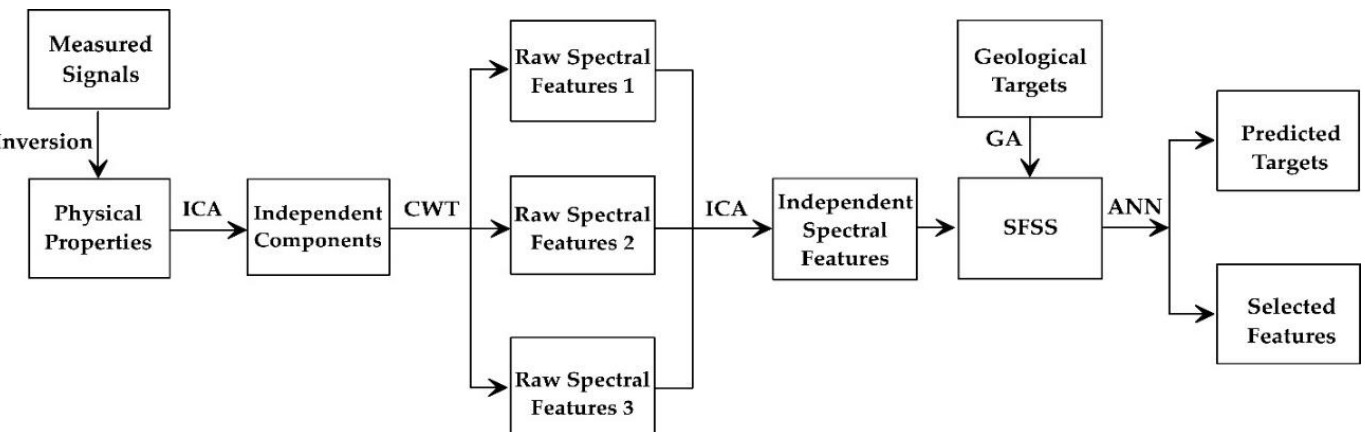

**Figure 1.** Workflow of SFSS for 3D predictive modeling.

### 2.1. Geological and Geophysical Settings

We test our method on an Au/Ag epithermal deposit known as the Newton deposit in British Colombia, Canada [28–30]. Regionally, the Late Cretaceous volcanic sequence

is overlain by Miocene–Pliocene Chilcotin Group flood basalts and Quaternary glacial deposits, which are variably eroded to expose the older rocks. Quaternary glacial tills cover most of the Newton property on a deposit scale. Consequently, deposit scale geological information has primarily been obtained from borehole samples [27–30]. A bedrock geology map of the property is compiled from the mapping of limited outcrops, drill cores, and cross-section interpretations (Figure 2a).

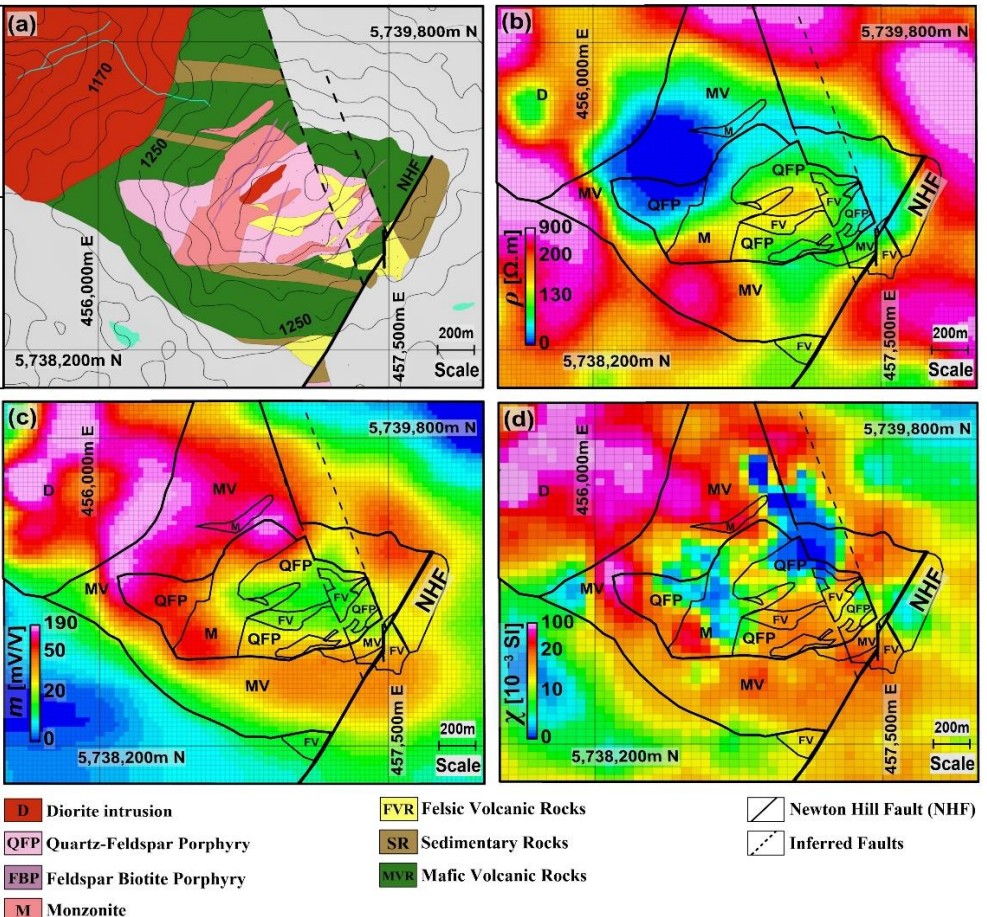

**Figure 2.** Geological and geophysical settings of the deposit: (**a**) Bedrock geology map at the deposit scale (sliced at an elevation of 1000 m). The light gray parts are undefined regions due to the lack of borehole information in the periphery of the deposit. Contours indicate topography. (**b**) Electrical resistivity model. (**c**) IP chargeability model. (**d**) Magnetic susceptibility model. The bedrock geology is overlaid on all geophysical models [27].

Three layered volcano-sedimentary sequences overlain from bottom to top are mafic volcanic rocks (MVR), sedimentary rocks (SR), and felsic volcanic rocks (FVR). The layered rocks are intruded by felsic to intermediate porphyritic intrusions, including the monzonite (M), quartz-feldspar porphyry (QFP), feldspar biotite porphyry (FBP), and the younger dioritic intrusion (D). The epithermal mineralization associated with the Newton deposit is at a depth of ~50 m to ~600 m. Au/Ag mineralization is mainly hosted within the felsic volcanic sequence [27–30]. The deposit is offset by the Newton Hill Fault (NHF), displacing geology and mineralization by ~300m of normal dip-slip movement.

Geophysical images of Newton deposit were obtained from 3D inversion of direct current (DC) resistivity, induced polarization (IP), and magnetic data sets [27,31]. In 2010, an 85 line-kilometer DC/IP survey was conducted in the Newton Hill area using a "pole-dipole" electrode configuration [27,31]. The DC/IP survey lines were placed at 200m intervals in the east–west direction in the hydrothermal alteration zone. The dipole

length was set to 100 m and 200 m, with a maximum of ten times dipolar separations. Apparent chargeability (mV/V) was measured by recording the voltage drop after current interruption. The total magnetic field data were collected from a helicopter flying at an average altitude of 155 m above the ground. The magnetic data were collected along N–S flight lines spaced 200 m apart, and E–W tie lines were flown approximately every 2000 m to 500 m. A total of 7071 line-kilometers of magnetic data were collected, covering an area of 1293 km$^2$. The magnetometer sampling rate was 0.1 s, yielding a measurement interval of approximately 10 m along each profile, depending on the helicopter speed [27,31].

The interrelationships between physical properties enabled the cooperative inversion of multiple datasets. The chargeability image in this study was used to constrain the magnetic inversion. The constrained susceptibilities set aside the less consistent equivalent solutions and narrowed down to a final solution that is well matched with the IP-DC resistivity inversion results, which improves deposit scale susceptibility imaging [27,31]. The value of this approach is that it does not necessarily require prior information for physical properties coupling. The total number of cells for forward DC/IP calculation is 288,000, and the size of each cell is 25 m × 50 m in the horizontal plane. The model cells are also set to grow exponentially with depth in 20 intervals up to 600 m. The unit cell size for cooperative susceptibility inversion is also set to X = 25 m, Y = 25, and Z = 10 m, with a total of 613,800 cells.

The recovered cooperative susceptibilities allowed to locate a magnetite destructive zone associated with porphyritic intrusions and felsic volcanoes (Au host rocks). Although, in some places, the distinction between magnetic highs of diorite and mafic volcanic rocks and magnetic lows of felsic volcanic rocks (Au host rocks) and porphyritic intrusions is still challenging. The result of the cooperative magnetic- IP-DC resistivity inversion is also shown in Figure 2. The high magnetic cap imaged from the constrained inversion is related to a mixture of the mafic volcanic rocks (MV) and the younger dioritic intrusions (D). A low magnetic zone (LMZ) was positioned deeper in earlier unconstrained models, but the cooperative inversion has replaced it upward (about 500 m) and imaged right under the high magnetic cap (magnetic susceptibilities > 0.04 SI). Several flanks of this LMZ locally reduce the magnetic susceptibilities to 0.015 SI. This structure is probably a response of the intermediate-to-felsic porphyritic intrusions (FIP) and interbeds of felsic volcanic rocks (FV) within the upper high magnetic cap [27,31].

### 2.2. Feature Extraction

The feature extraction in this study is based on implementing several blind source separation methods, including PCA, ICA, and CWT, to recover latent features from a set of highly mixed images. Consider a mixing model where every image (g) is a mix of several hidden features (f) with different contributions to constructing the observed image, determined by a set of mixing weights *A*, in the form of a linear mixing model [18]:

$$g = Af \tag{1}$$

To recover the hidden features (f), one needs to find a separation matrix (*W*) and unmixes the observed images:

$$f = Wg = A^{-1} g \tag{2}$$

The separation matrix (*W*) can be estimated as an optimization problem by minimization of a cost function. By making a few assumptions about the statistical measures of the data sets, the feature extraction process iteratively reduces the effect of the mixing on the observed images. In PCA, the second-order statistical measure, i.e., variance, is maximized for image separation, and the outcome is linearly separated uncorrelated images. However, observed images with the nonlinear form of correlation (dependency) pose a significant difficulty to PCA-based separation methods. We can solve this problem by maximization of higher-order statistical measures (non-gaussianity) to separate the images into nonlinearly uncorrelated images through ICA.

The ICA starts with a PCA as a preprocessing step that removes the mean of the input images (g). The principal components (y) are related to the centered images ($g_c$) through a weight matrix *D*:

$$y = Dg_c \tag{3}$$

The matrix *D* can be estimated through variance maximization of the principal components (y). Finally, increasing the non-gaussianity of the principal components (y) produces the ICs of the images. The problem is to find a rotation matrix that, during the multiplication with the principal components, produces the least Gaussian outputs [16,18].

The Fast-ICA algorithm, through negentropy maximization and the Hyvärinen fixed-point method, was used to obtain the rotation matrix that incorporates higher-order statistics to recover the latent independent sources [16,18]. The entropy (*H*) of an image (g) is defined as [16,18]:

$$H(g) = - \int p(g) \log p(g) dg \tag{4}$$

where $p(g)$ is the probability density of the image g. Entropy is a measure of randomness. The more unpredictable and unstructured the variable is, the larger its entropy. Theoretically, the Gaussian variable possesses the largest entropy. The negentropy (*J*) of an image is the normalized differential entropy of that image:

$$J(g) = H\left(g_{gausss}\right) - H(g) \tag{5}$$

where $H(g)$ is the entropy of the image, $H\left(g_{gausss}\right)$ is the entropy of a Gaussian random image of the same covariance matrix, and $neg(g) \geq 0$ is always non-negative and zero when the image has a pure Gaussian distribution. The negentropy maximization is based on bringing an objective function to an approximated maximum value [14,16–18,27].

The CWT in two dimensions (horizontal planes) was used for feature extraction in the frequency domain. Since wavelets are localized in space and have finite durations, the sharp changes in images are efficiently detectable by 2D wavelet decomposition, which provides a unique way for spectral feature extraction [19,32]. The output of wavelet decomposition effectively reflects the sharp changes in images, making it an ideal tool for feature extraction [19,32]. The continuous wavelet transform of an image *I (x, y)* is defined as a decomposition of that image by translation and dilation of a mother wavelet *ψ (x, y)*. The resulting wavelet coefficients ($C_s$) are then given by

$$C_S = (b_1, b_2, a) = \frac{1}{\sqrt{|a|}} \iint I(x,y) \psi^* \left( \frac{x - b_1}{a}, \frac{y - b_2}{a} \right) dx dy \tag{6}$$

where $b_1$ and $b_2$ control the spatial translation, $a > 1$ denotes the scale, and $\psi^*$ is the complex conjugate of the mother wavelet *ψ (x, y)*. The mother wavelet shifts and scales in multiple directions and produces numerous features. In this study, taking advantage of the nICA statistical properties, we can keep the most geologically pertinent information within the spectral decomposed volumes [18].

### 2.3. Learning Spectral Features

Spatial and spectral feature extraction provides inputs for predictive modeling through supervised machine learning algorithms. Training machine learning models directly with raw data sets often yield unsatisfactory results. The feature extraction process identifies the most discriminating characteristics in raw images, which a machine learning algorithm can more easily consume. However, the curse of dimensionality prevents efficient machine learning when extracted features are too large [8–10]. Anyway, dimensionality reduction can improve spectral learning performance through feature extraction and feature subset selection methods. The main difference is that feature extraction combines the original features and creates a set of new features, while feature selection selects a subset of the original features [15,21–23,26]. We propose a hybrid algorithm that includes several phases

of preprocessing, spatial feature extraction, spectral feature extraction, and machine learning for geophysical predictive modeling (Figure 3). In this section, we first outline a basis for SRL through multilayer perceptron (MLP), and then we describe our proposed SFSS algorithm for 3D geophysical predictive modeling, feature selection, and dimensionality reduction.

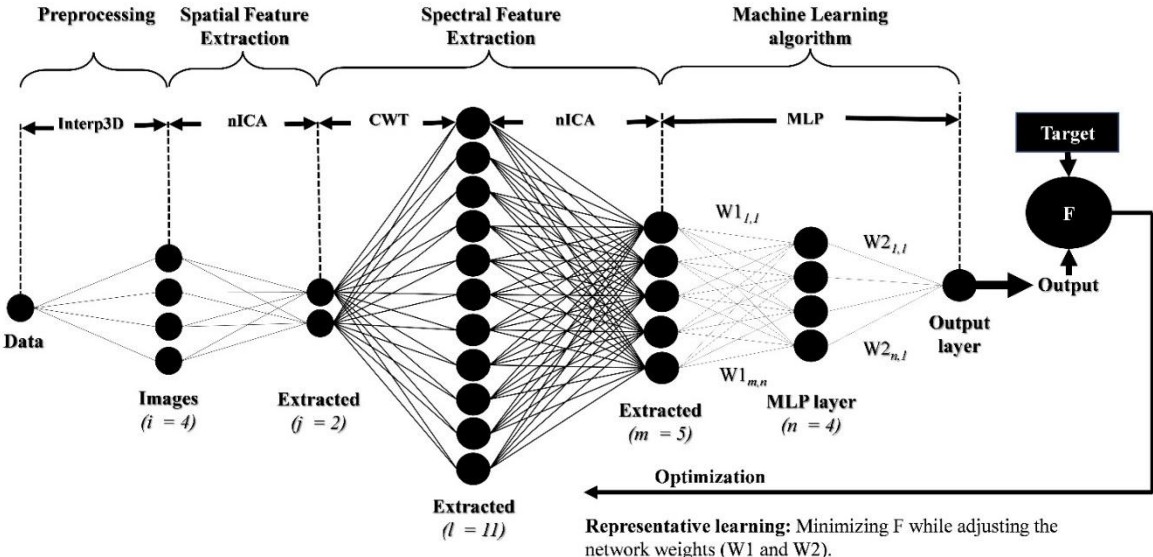

**Figure 3.** Schematic view of SRL for predictive modeling. The algorithm consists of four sub-modules: The preprocessing prepares the data sets by 3D inversion, interpolation, and filtering methods. The spatial feature extraction with nICA separates the image overlaps in 3D. Spectral feature extraction with CWT-nICA extracts the wavelet decomposed features. Furthermore, the machine learning algorithm with MLP adjusts network weights (W1 and W2) to learn patterns inside the extracted features based on the sample targets.

The basic building block of MLP is the Perceptron, a mathematical analog of the biological neuron, first described by Rosenblatt [33]. In this model, the integrated weights and biases of input vectors (*x*) are activated by a sigmoid function (activation function) to produce the output (*y*). After setting the initial weights randomly, the network is ready to train. Then, the output will be compared to the target in an iterative process, adjusting the weights of the ANN. This process can be performed by defining an Objective Function (*F*) and minimizing it over N training samples. *F* is defined as the least mean square (LMS) of the output vector ($y_i$) and the desired output or the target vector ($t_i$):

$$F = \frac{1}{N} \sum_{i=1}^{N} (t_i - y_i)^2 \tag{7}$$

During the back-propagation process, the optimization updates the synaptic weights to make output closer to the specified target.

Despite successful applications of ANNs in nearly all branches of science and engineering, we still face many problems and potential pitfalls. A potential pitfall is an over-fitting problem, in which designing too many neurons and too many iterations gives rise to noisy outputs. On the other side, under-fitting happens when a too-simple network is created, sometimes leading to under-estimated values. Preprocessing of original data is also an important criterion. Random noises in inputs or targets can propagate through networks and disturb the outputs. For example, surficial noises in geophysical images can lead to underestimating the results. Smoothing the inputs and targets can reduce spurious effects but also eliminate valuable attributes in the images, leading to overestimation in the final

models. An optimum smoothness filtering strategy can be used to avoid over-fitting and under-fitting effects [34–36].

Choosing an efficient optimization algorithm is also critical because a suitable optimization algorithm can make a considerable difference in final estimations. The Levenberg–Marquardt optimization technique [37,38] is a fast and robust technique for minimizing the performance function. Because of the local minima problem inherent in nonlinear optimization procedures, finding a global optimal or better local solution is the goal of an effective learning process. Several novel heuristic stochastic techniques are available to solve this complex hyper-dimensional optimization problem [24,35,38–42].

In this study, we use GA optimization to minimize the ANN cost function and the number of input features simultaneously. The process is called feature subset selection and begins with initializing the input parameters we wish to optimize. These considerations are codified in an objective or cost function with different weights depending on the survivability of individuals. The best-fitted individuals meet the specifications of the objective function and survive, while the rest of the population is extinct [24,41,42]. The optimal solution emerges from parallel processing among the population with a complex fitness landscape. The whole idea is moving the population away from local optima that classical hill-climbing techniques usually might be trapped in. This ability makes GA an extremely robust global optimization technique that can resolve the traditional problem of local pockets existing in older optimization routines.

During the SFSS procedure, the spectral extracted features are fed to the MLP network, and the learning process involves adjusting the network weights and the number of the selected features while minimizing the bi-objective function F (Figure 4). In this study, we used a fast implementation of the Non-dominated Sorting Genetic Algorithm (NSGA) for bi-objective optimization. The NSGA algorithm [43] optimizes a global bi-objective cost function in the general form of:

$$E_{\text{Global}} = f\left(E_{\text{ANN}}, n_f\right) \tag{8}$$

where $f$ is the global cost function operator. The bi-objective GA simultaneously minimizes the number of spectral features ($n_f$) and ANN cost ($E_{\text{ANN}}$).

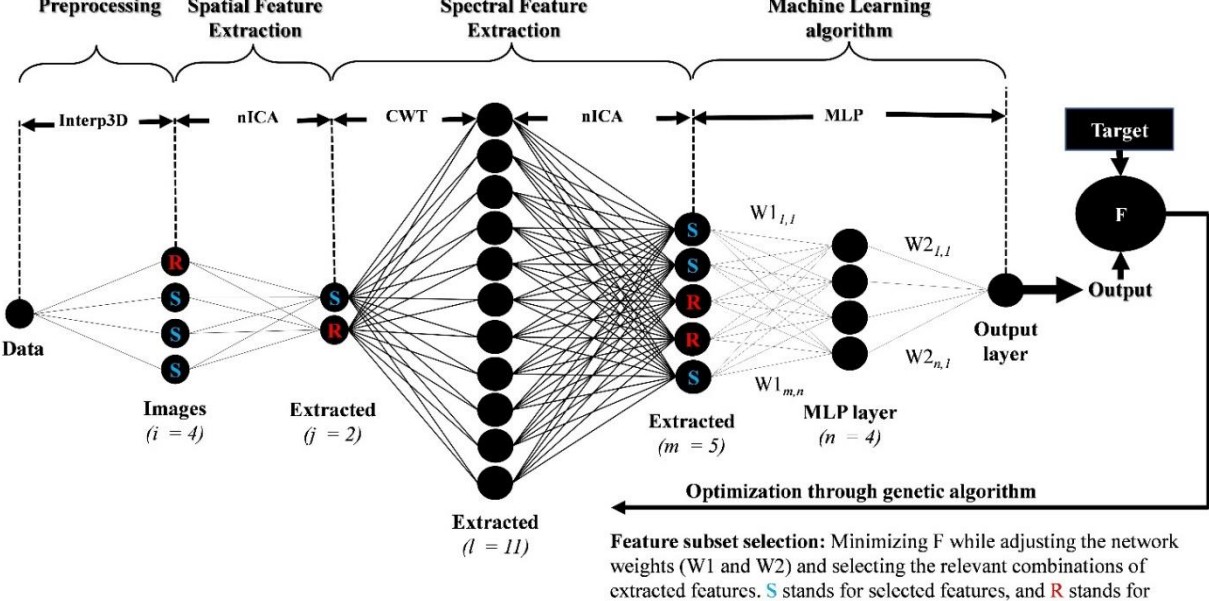

**Figure 4.** Schematic view of spectral feature subset selection for predictive modeling. The algorithm consists of four main sub-routines like the SRL algorithm, except that the MLP is integrated with NSGA-II to adjust the network weights and the selection of inputs simultaneously.

The solution of multi-objective optimization problems gives rise to a set of Pareto-optimal solutions instead of a single solution. Conventional optimization methods convert multi-objective optimization problems to single-objective optimization problems by emphasizing one Pareto-optimal solution at a time. The algorithm has to be iterated to obtain a different solution at each run to find multiple solutions. Over the past decades, evolutionary algorithms have been suggested to find multiple Pareto-optimal solutions in one single simulation run, emphasizing moving toward the actual Pareto-optimal region. This study used an improved version of NSGA called NSGA-II [44,45]. NSGA-II can be detailed in the following steps:

*Step 1: Population initialization*

NSGA-II randomly initializes a population $P_t$ of size $N$. An offspring population $Q_t$ is generated using the genetic operators (tournament selection, crossover, and mutation), and $P_t$ and $Q_t$ are combined to create a population $R_t$ of size $2N$.

*Step 2: Fast non-dominated sorting procedure*

The algorithm sorts the combined population $R_t$ according to non-domination to obtain different non-dominated fronts $F_i$.

*Step 3: Crowding distance estimation*

To maintain the population diversity in each non-dominated front, the crowding distance is computed; that is, the average distance between two points on either side of a particular solution. Then, repeat the process with other objective functions.

*Step 4: Binary tournament selection*

Individuals are selected using a binary tournament selection with a crowded comparison operator to create a mating pool: The best solutions in the combined population are those belonging to the lower-level non-dominated set $F_1$. When the two solutions have an equal non-domination level, the solution with a higher crowding distance is preferred to preserve diversity.

*Step 5: Crossover and mutation*

The algorithm creates an offspring population from the mating pool by applying the simulated binary crossover operator and the polynomial mutation operator.

*Step 6: Recombination and selection*

The last stage of the NSGA-II involves combining the current population with the offspring and then selecting the best solutions from this combined population to create a new population for the next generation. The process is iterated for the desired generations to achieve the best multi-objective solution in the last generation.

## 3. Results and Discussion

We compiled several MATLAB functions and scripts to predict geological targets in 3D, such as 3D lithotype estimation and 3D Au grade estimation. We used unevenly distributed borehole datasets, including the lithotype data (Figure 5a) and the Au concentrations in g/t (Figure 5d). Borehole datasets show that the hydrothermal alteration has destroyed magnetite and replaced it with pyrite in the felsic volcanic rocks that are the host rocks of epithermal Au/Ag mineralization [27–30]. Therefore, the high-Au grades are accompanied by low-magnetic anomalies of the felsic rocks. This correlation made it possible to separate the high magnetics of the mafic volcanic rocks and dioritic intrusions from low-magnetic felsic volcanic rock and porphyritic intrusions based on the 3D cooperatively recovered magnetic susceptibility model.

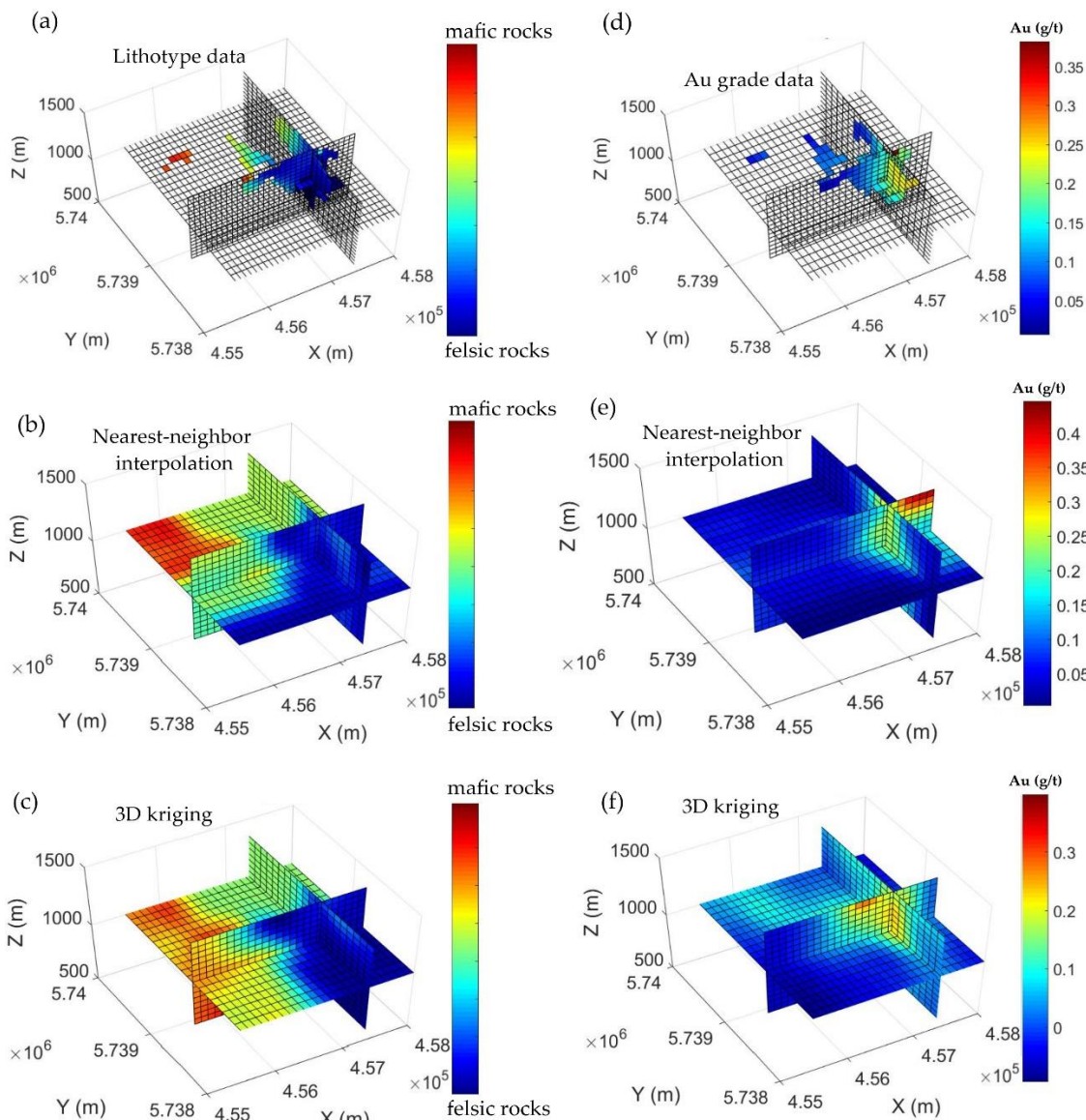

**Figure 5.** Interpolation of geological targets with conventional methods. (**a**) Lithotype data (mafic/felsic). (**b**) Nearest-neighbor interpolation of lithotypes. (**c**) Kriging of lithotypes. (**d**) Au grade data. (**e**) Nearest-neighbor interpolation of Au grades. (**f**) Kriging of Au grades [27].

Consequently, we assigned three lithotype codes to quantify the lithological variations: Code 1 for felsic volcanic rocks, code 2 for mafic volcanic rocks, and code 3 for dioritic intrusions. The lithotype codes are matched and sorted according to their magnetic properties, from low magnetics of felsic volcanic rocks and intrusions to high magnetics of mafic volcanic rocks and diorites. The datasets are interpolated with the nearest neighbor (direct gridding) and kriging methods for comparison. Figure 5 shows that traditional interpolation methods cannot add any new geological information in places without borehole data. This study aims to solve this 3D interpolation problem by reconstructing lithological and Au grade patterns from 3D geophysical images.

The 3D physical property images are derived from cooperative inversion [27,31] of magnetic and DC/IP datasets (Figure 2b–d). A preliminary Fast-ICA separates the latent features inside the multiple physical property images in the form of three negentropy-maximized ICs (spatial feature extraction). Then, three ICs are decomposed to form a set of raw spectral features through a continuous wavelet transform (CWT).

We calculated the 2D wavelet coefficients with a gaussian mother wavelet for three scales in 72 directions (every 5°). We used only three scales because we observed that for more than three scales, the changes in low-frequency features (high scales) would disappear, and there will not be any difference in the raw spectral features for scales more than three. The 3D spectral decomposition is iterated for the three initial ICs and has produced 648 (3 × 72 × 3) features containing several frequency-dependent raw features that need further extraction. Figure 6 demonstrates the raw spectral features sliced at an elevation of 1000 m. X and Y UTM coordinates are removed to save space for display.

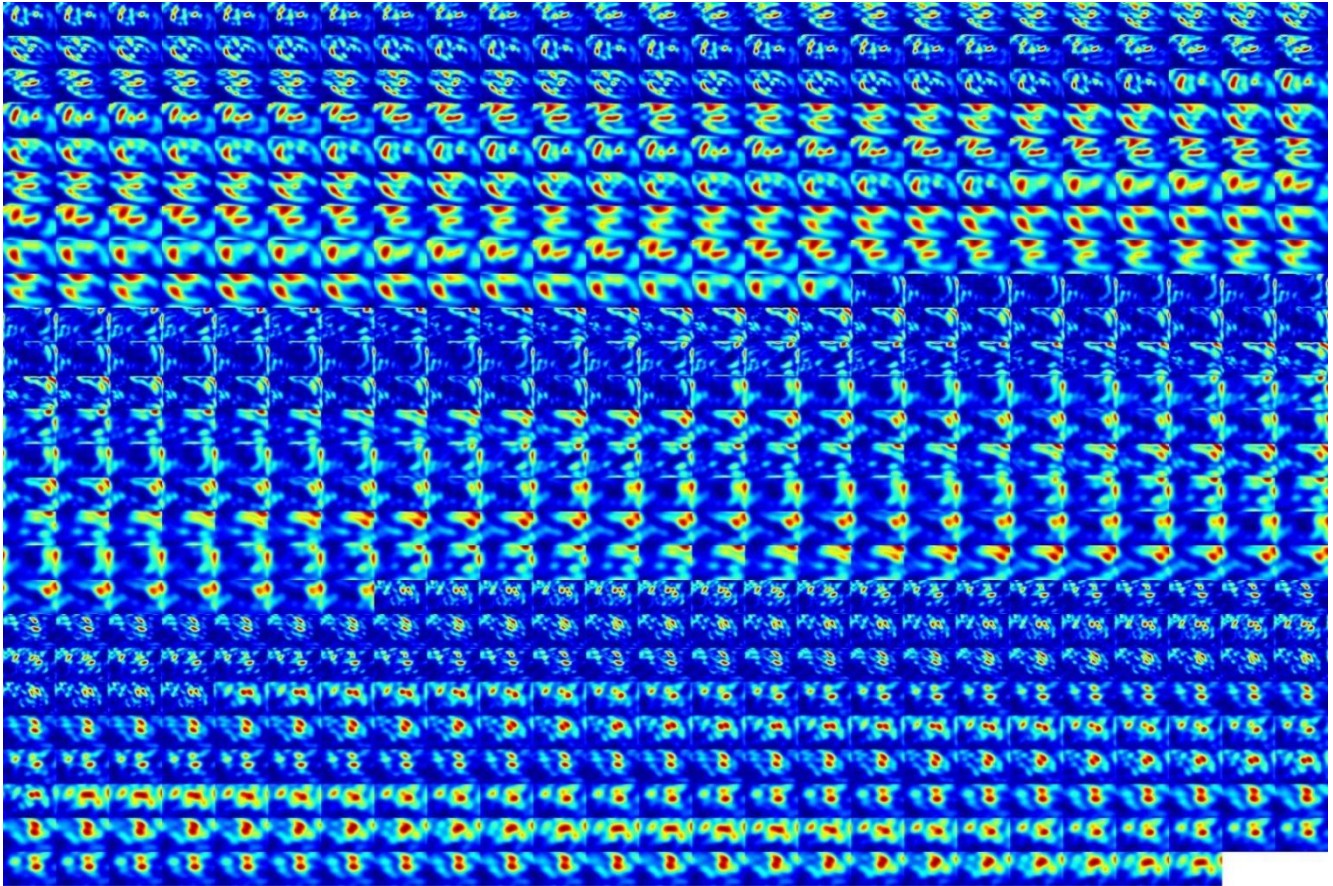

**Figure 6.** CWT with a gaussian mother wavelet produced raw 3D spectral features with three scales in 72 directions (0°, 5°, 10°, . . . , 355°). The extracted features are sliced at an elevation of 1000 m.

Fast-ICA, through negentropy maximization, separates the 648 raw features to produce the 3D independent spectral inputs necessary for feature selection. The Fast-ICA algorithm can also reduce the dimensionality of the raw features. The best strategy to reduce the dimensionality of features depends on two factors: computer hardware resources and the stability of the SFSS output. Low reductions result in high computational costs, and severe reductions lead to loss of valuable information and poor predictions. There is always an optimum way to moderately reduce the dimensionality of features through several testing and running of the algorithm by the user. In cases with no hardware limitation, this trial-and-error strategy is unnecessary. In this study, we reduced the dimensionality of the raw spectral features by 35 percent to 227 extracted spectral features (Figure 7).

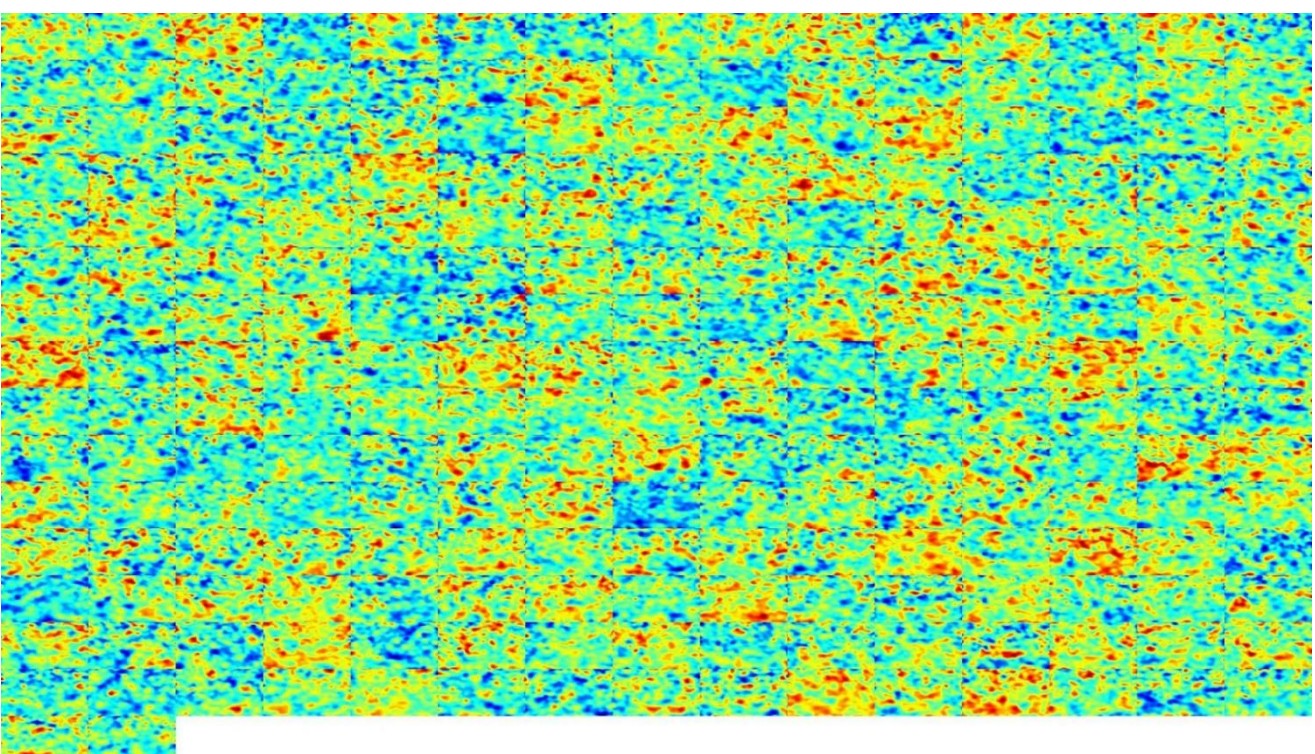

**Figure 7.** Total 227 number of separated 3D spectral features after Fast-ICA dimensionality reduction for lithological feature selection. The extracted features are sliced at an elevation of 1000 m.

We ran the SFSS algorithm two times in parallel to predict lithotypes and Au grades. For feature selection, we used an MLP network with independent spectral features as inputs, 20 neurons in the hidden layer, a maximum of 20 iterations, and one output. Of the borehole data, 70% are used for training and the rest for validation (10%) and testing (20%). The GA is iterated in 20 generations with 20 populations, 70% crossover, 40% of mutation, and a mutation rate of 0.05.

For the lithotypes model reconstruction, evaluation results are compared for both SRL optimized with the Levenberg–Marquardt method and the SFSS algorithm optimized with GA (Figure 8a). As can be seen, the SFSS algorithm gives better validation and test results (98% and 91% R-squares, relatively) compared to the poor results of the conventional SRL method (with 90% and 27% R-squares, relatively).

For Au grade prediction, the same network parameters are used, and the Fast-ICA dimensionality reduction has reduced the number of independent spectral features to 227 (Figure 7). Evaluation results are compared for both SRL optimized with the Levenberg–Marquardt method and the SFSS algorithm optimized with GA (Figure 8b). As can be seen, the SFSS algorithm gives much better validation and test results (with 97% and 88% R-squares, relatively) compared to the poor results of the conventional SRL method (with 95% and 56% R-squares, relatively).

The error landscapes of the lithotype and Au grade prediction by SFSS are also shown in Figure 9. The best results come after the last generation of GA and for the least global error indicated by the dark blue color. For lithotype prediction, the SFSS algorithm selects 200 out of 227 features for spectral learning (Figure 10). As can be seen, the SFSS can recover a much better lithological model compatible with prior geological information. The selected features for lithotype prediction are also shown in Figure 11. In this case, SFSS detects 27 redundant features in the prediction of lithological variations that helps to facilitate the machine learning predictions.

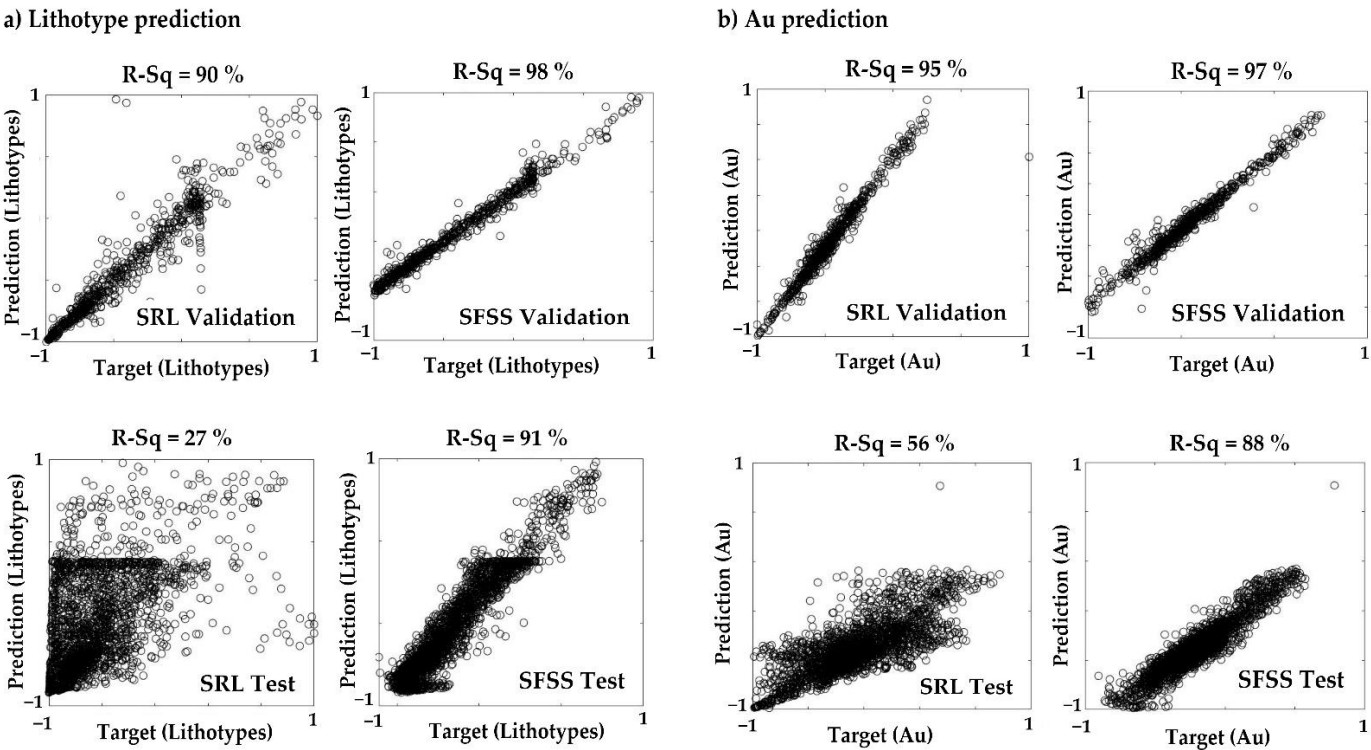

**Figure 8.** Evaluation of lithotype (**a**) and Au (**b**) predictive modeling for SRL and SFSS algorithms. R-sq denotes the R-squared values.

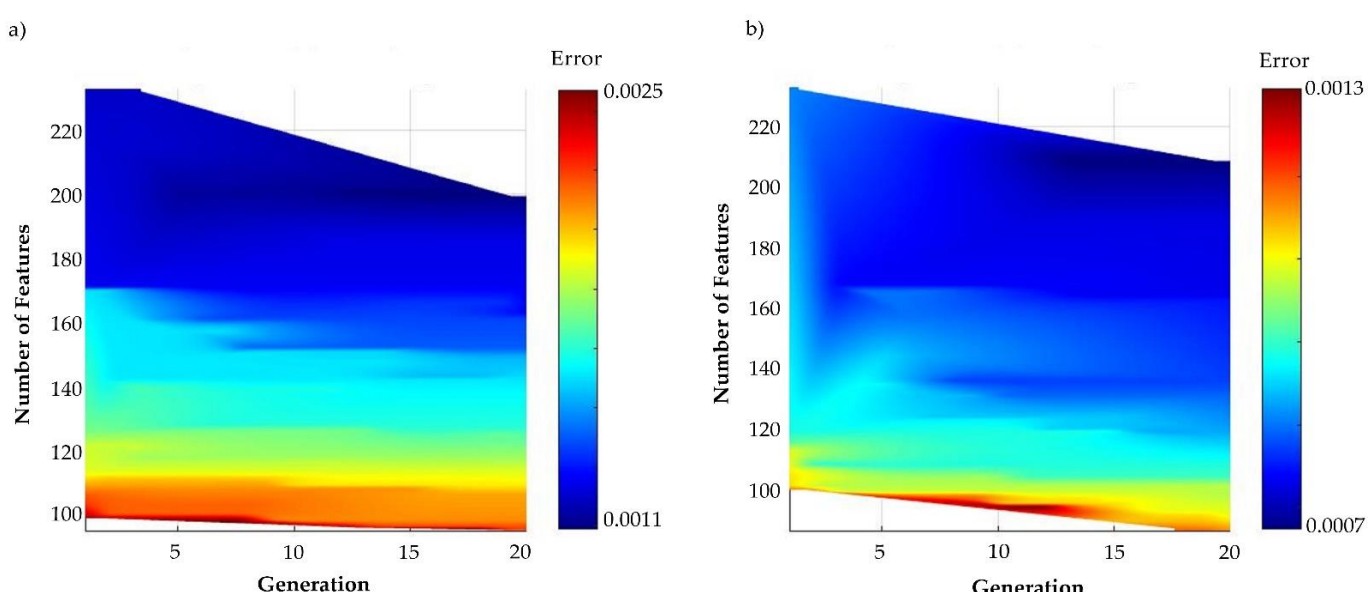

**Figure 9.** Bi-objective error landscape of SFSS algorithm during lithotype (**a**) and Au (**b**) estimation.

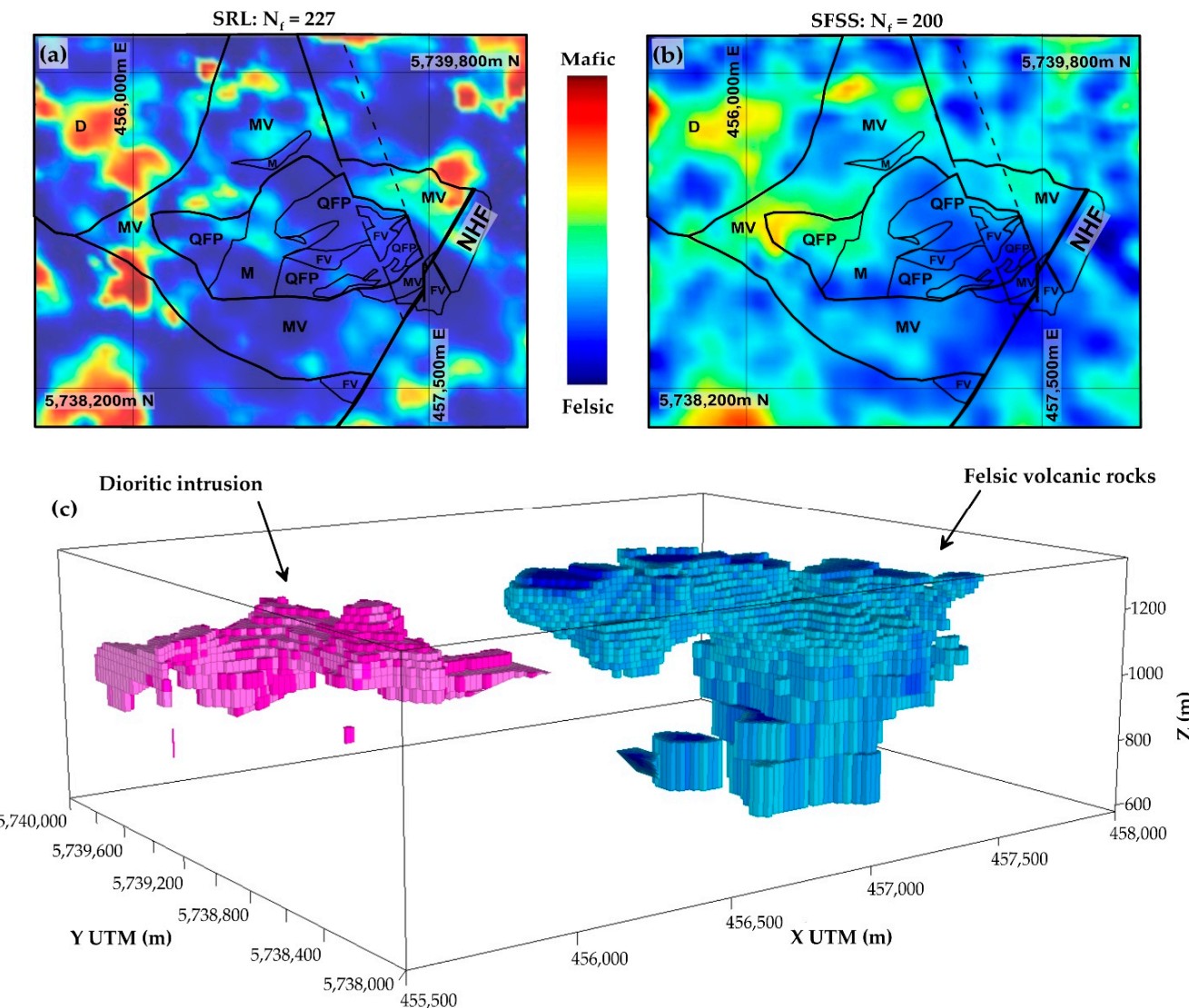

**Figure 10.** Results of SRL lithotype predictions (**a**) versus SFSS lithotype predictions (**b**,**c**). SFSS resulted in the better reconstruction of lithotypes with a smaller number of input spectral features (Nf = 200). Results are sliced horizontally at the elevation of 1000 m.

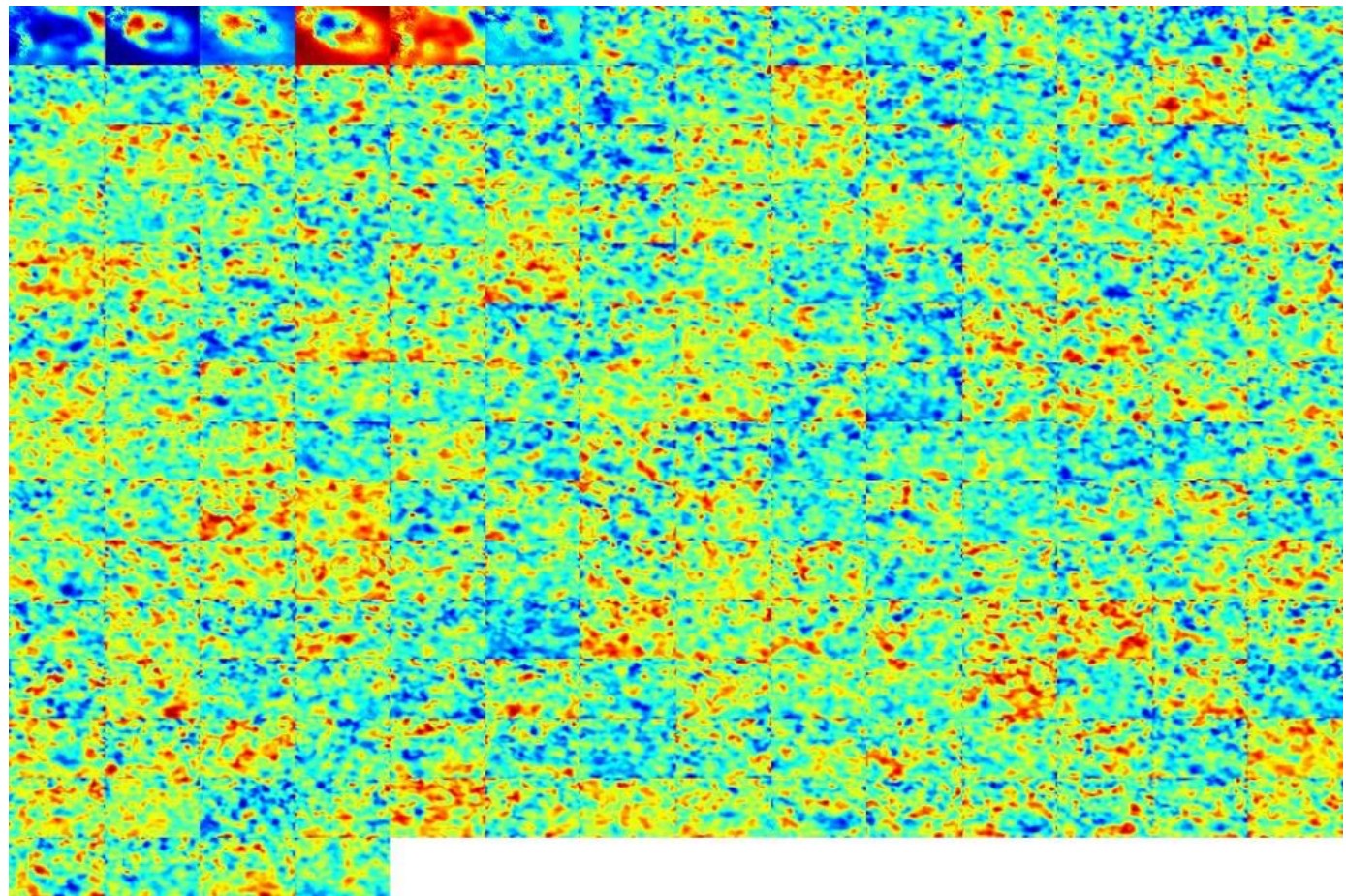

**Figure 11.** A total of 200 selected features were used for lithotype prediction. The selected features are sliced at an elevation of 1000 m.

The error landscape of the Au grade prediction by SFSS is also shown in Figure 10b. For the last generation, the SFSS algorithm selects 209 out of 227 features for spectral learning (Figure 12). The SFSS can recover a much better Au grade model than conventional SRL estimations. The selected features are also shown in Figure 13. In this case, SFSS detects 18 extremely redundant features in the prediction of Au concentrations that help to facilitate the machine learning predictions.

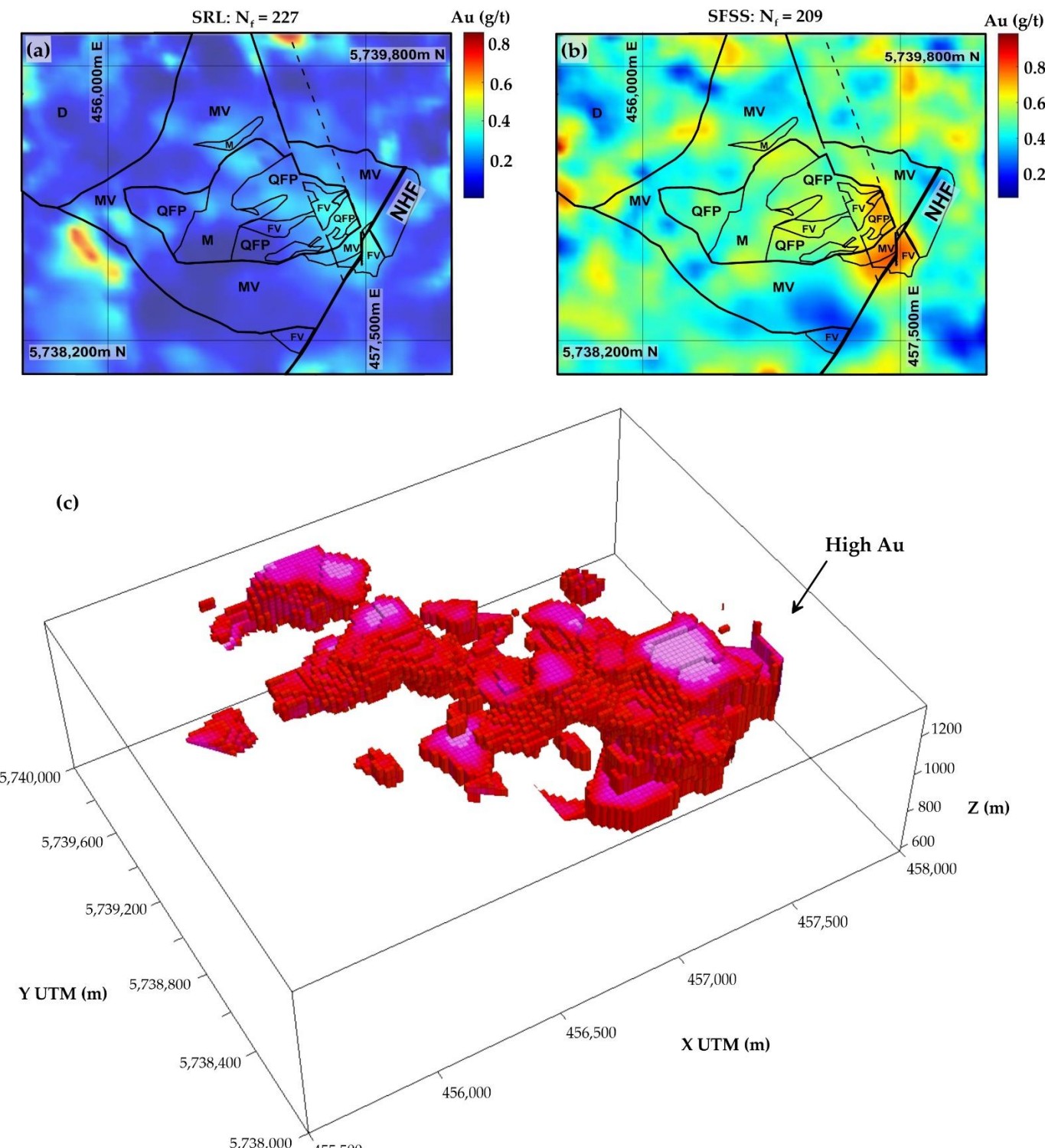

**Figure 12.** Au grade predictions: Results of 3D SRL predictions (**a**) versus 3D SFSS predictions (**b**,**c**). SFSS resulted in the better reconstruction of Au-grade distributions with a smaller number of input spectral features ($N_f$). Results are sliced horizontally at the elevation of 1000 m.

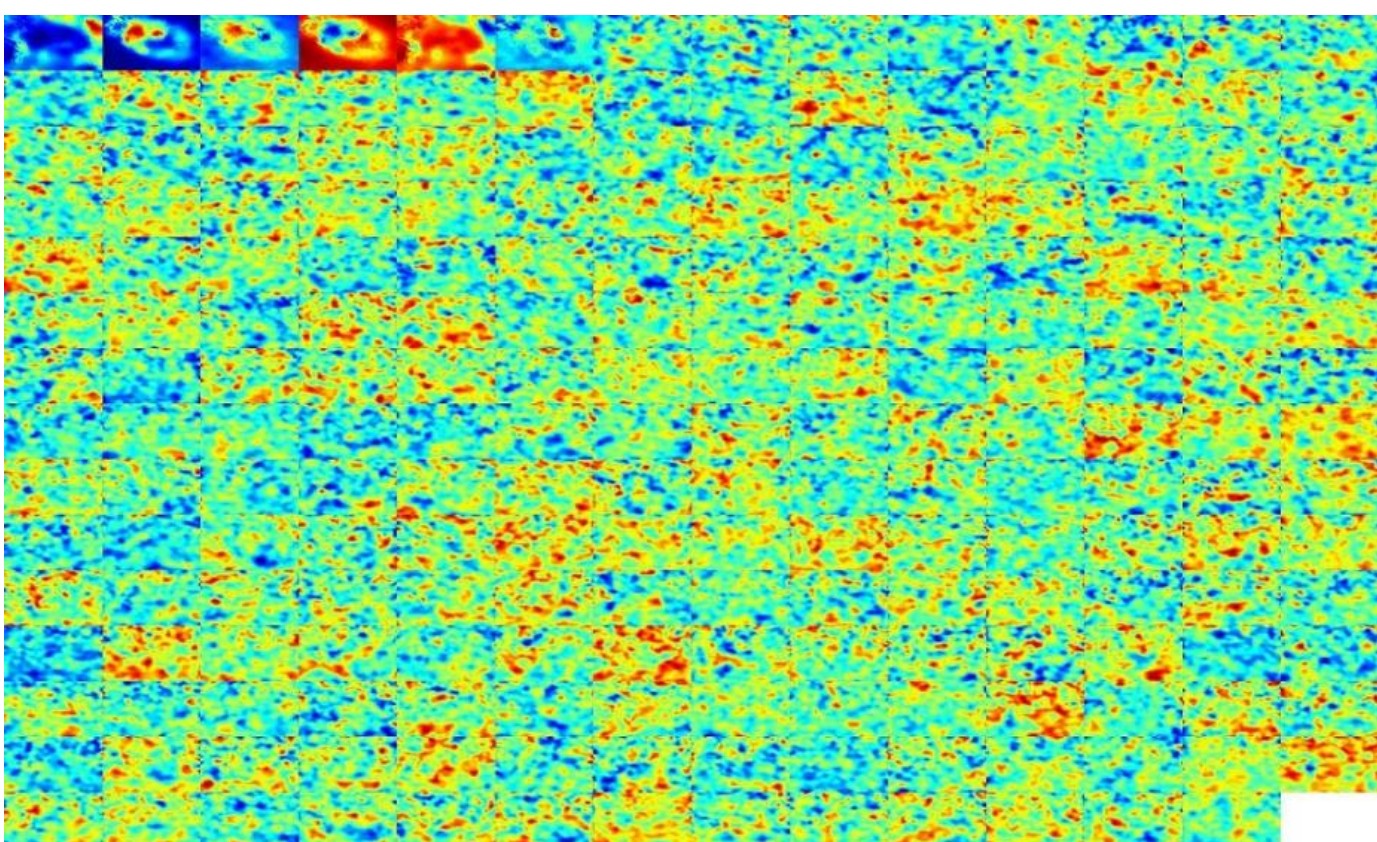

**Figure 13.** A total of 209 selected features were used for Au prediction. The selected features are sliced at an elevation of 1000 m.

## 4. Conclusions

We propose a spectral feature selection algorithm for supervised learning of geological patterns to perform 3D predictive modeling from 3D inverted physical properties. Our work is based on the synergy between the ICA and CWT feature extraction methods and multi-objective machine learning optimization through GA for feature selection. The spectral feature extraction method provided the inputs of the feature selection algorithm, and we show that our self-proposed SFSS algorithm can pick the relevant spectral features necessary for 3D predictive modeling of the targets. The practical implementation of the SFSS algorithm is also evaluated for an epithermal Au/Ag deposit in British Columbia, Canada. The results show that the spectral learning scheme proposed can efficiently learn geological patterns to make predictions based on 3D physical property inputs. The SFSS also minimizes the number of extracted spectral features and tries to pick the best representative geophysical features for each target learning case. This automated dimensionality reduction strategy also gives interpreters a precise predictive model and an understanding of the relevant and irrelevant selected geological features at the end, which adds value to the interpretability of the machine learning process. Although tested on Newton's epithermal deposit, the proposed feature selection approach should be applicable to similar mineral deposits. Other 3D geophysical imageries acquired with inversion of seismic, gravity, and electromagnetic data sets could be considered as well to enhance the accuracy of the feature extraction and the subsequent feature selection. Future research will be focused on the automatic fine-tuning of SFSS hyperparameters and adapting the 3D SFSS algorithm to provide a practical tool for integrated 3D imaging of mineral deposits.

**Author Contributions:** Conceptualization, B.A. and L.-Z.C.; methodology, B.A.; software, B.A.; validation, B.A. and L.-Z.C.; formal analysis, B.A.; investigation, B.A.; resources, B.A.; data curation, B.A.; writing—original draft preparation, B.A.; writing—review and editing, B.A., L.-Z.C., M.J., and D.L.; visualization, B.A.; supervision, L.-Z.C.; project administration, L.-Z.C.; funding acquisition, L.-Z.C. All authors have read and agreed to the published version of the manuscript.

**Funding:** This study was supported by grants from the Natural Sciences and Engineering Research Council of Canada (Grant No. 43505) and Fonds de recherche du Québec—Nature et technologies (Grant No. 133896).

**Data Availability Statement:** Not applicable.

**Conflicts of Interest:** The authors declare no conflict of interest.

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
