# Peer review of "3D Geophysical Predictive Modeling by Spectral Feature Subset Selection in Mineral Exploration"

_minerals, doi:10.3390/min12101296_

Round 1
Reviewer 1 Report
The paper presents an algorithm for prediction of 3D lithology from 3D geophysical property images. The algorithm is featured by spectral feature learning that utilizes independent component analysis, continuous wavelet transform, and artificial neural networks. The proposed method was validated in a case study in the Newton deposit. The manuscript is general well-written, but minor revision is required to consolidate the work before it can be accepted.
The title should reflect that the algorithm makes predictions from 3D geophysical images.
Equation (9). Please present the detailed form of the global cost function operator.
The GA algorithm generally requires many iterations to find the minimum. Please present timing and iterations for solution of Equation (9). There are other algorithms such as Baysian optimization, which is deemed can be more efficient in optimimizing n_f in Equation (9).
Shahriari, B., Swersky, K., Wang, Z., Adams, R. P., & De Freitas, N. (2015). Taking the human out of the loop: A review of Bayesian optimization. Proceedings of the IEEE, 104(1), 148-175.
To extract features and make predictions, maybe the reader can compare and discuss other deep learning methods such 3D CNN.
Overall, I am looking forward to a richer final version of this paper.
Author Response
Point 1: The title should reflect that the algorithm makes predictions from 3D geophysical images.
Response 1: The title is modified to 3D geophysical predictive modeling by spectral feature subset selection in mineral exploration.
Point 2: Equation (9). Please present the detailed form of the global cost function operator.
Response 2: The global cost function operator is mentioned to show the general form of the optimization problem. Further details are explained later, followed by NSGA-II algorithms’ steps.
Point 3: The GA algorithm generally requires many iterations to find the minimum. Please present timing and iterations for solution of Equation (9). There are other algorithms such as Baysian optimization, which is deemed can be more efficient in optimimizing n_f in Equation (9).
Response 3: It would be an interesting study to compare the NSGA-II algorithm to a Bayesian optimization approach. In this study, we were not concerned about the convergence speed and number of generations/iterations. However, the number of GA generations and ANN iterations are detailed in the discussion section. Typically, on an average high-performance computer, with a 6-core CPU (Core i7) and 5GHz clock speed, and 64 Gb RAM, it takes several days to complete a run. We can also rewrite the codes to accelerate it with GPU integration. Nevertheless, it is a whole new study. We are planning to do that in the near future.
Point 4: To extract features and make predictions, maybe the reader can compare and discuss other deep learning methods such 3D CNN.
Response 4: The SFSS algorithm is indeed very much similar to CNN algorithms, except that it lacks a few preprocessing techniques like average pooling, max pooling, ReLU activation, SoftMax, etc.

Reviewer 2 Report
The work is very interesting, of great applicability and quite complex. However, there are some concepts that are a little confusing in the manuscript and that should be clarified, namely:
· The abstract needs slight improvements.
· The Introduction section should finish with the main objective and the specific objectives of the work.
· Section 2- the authors should clarify if the method proposed (3D SFSS) is self-proposed.
· Section 2- steps 2 and 4 are not simultaneous? (before figure 1)
· L178-181: from my point of view, it is not necessary to make this comparison with PCA...
· L191-195: some clarifications are needed.
· In the Conclusions Section include the work limitations and the applicability for other areas with different geological/geophysical conditions
See all my comments in the attached PDF file.

Author Response
Point 1: The abstract needs slight improvements.
Response 1: The abstract is modified entirely based on other reviews.
Point 2: The Introduction section should finish with the main objective and the specific objectives of the work.
Response 2: Changes made according to the review.
Point 3: Section 2- the authors should clarify if the method proposed (3D SFSS) is self-proposed.
Response 3: Changes made according to the review.
Point 4: Section 2- steps 2 and 4 are not simultaneous? (before figure 1)
Response 4: Steps 2 and 4 are different. Step 2 is for Spatial Feature Extraction. Step 4 comes after and works as spectral feature extraction.
Point 5: L178-181: from my point of view, it is not necessary to make this comparison with PCA.
Response 5: Changes made according to the review.
Point 6: L191-195: some clarifications are needed.
Response 6: Changes made according to the review.
Point 7: In the Conclusions Section include the work limitations and the applicability for other areas with different geological/geophysical conditions
Response 7: Changes made according to the review.

Reviewer 3 Report
This paper lacks cohesion. It seems like different sections have been written by different authors without an effort to homogenise these sections. The introduction and the abstract lack a clear description of the objective of the methodology presented. The title is misleading as predictive modelling is different from geophysical integration, which is the topic of the paper. Secondly, geophysical integration is seeing as a matter of integrating images, but it is more complicated than that. First, geophysical images, particularly for potential field methods, are affected by the non-unicity of the solutions, particularly for unconstrained inversion. The second point, which is particularly discussed in the description of your field case, is related to physical properties, with are not considered in your methodology. In conclusion, your introduction should include a solid discussion on building geological targets from geophysical data, with a breakdown on the steps you want to address in your paper. Do not neglect to describe the previous steps accomplished on a data set because they might have an impact on what you want to achieve.

Author Response
Point 1: The introduction and the abstract lack a clear description of the objective of the methodology presented.
Response 1: The introduction and abstract are modified based on all reviews.
Point 2: The title is misleading…
Response 2: Changes made according to the review.
Point 3: Are you trying to recover the rules of your cooperative inversion?.
Response 3: We try to automatically interpret the geophysical 3D images through feature extraction and predictive modeling by feature selection.
The estimation of true depth in 3D inversion is very important, especially for deeper targets. A poor unconstrained inversion results in poor and sometimes misleading 3D predictive models at the end. That's why we used a cooperative approach for inverse modeling. A reference is cited as a chapter of my Ph.D. thesis defended in 2018:
[27. Abbassi, B. Integrated imaging through 3D geophysical inversion, multivariate feature extraction, and spectral feature selection. PhD Thesis, Université du Québec à Montréal and Université du Québec en AbitibiTémiscamingue, 2018].
About the effect of 3D inversion and inversion parameter initialization on feature extraction, we also published a paper in 2021:
[17. Abbassi, B.; Cheng, L.-Z. 3D Geophysical Post-Inversion Feature Extraction for Mineral Exploration through Fast-ICA. Minerals 2021, 11, 21, doi:10.3390/min11090959.].

Round 2
Reviewer 2 Report
The authors improved the manuscript according to the reviewers’ suggestions, namely:
The title was changed.
The abstract was improved.
The main objective of the work was added in the Introduction Section
In Section 2 the authors clarify that the method proposed is self-proposed.
The steps in Figure 1 were clarified.
The importance of PCA in this approach was clarified.
In the Conclusions Section were included the work limitations and the applicability for other areas.
In addition to these aspects, other parts of the manuscript were improved and new parts were added.
Author Response
Thanks for the comments. I assume that there is no question or remark to be answered.
Reviewer 3 Report
See file attached.

Author Response
Point 1: Are you using your results for predicting Au-grade? If yes, this should be mentionned in your abstract and more clearly in your paper.
Response 1: Changes made according to the review.
Point 2: Is the top part of this Figure repeated?
Response 2: No it is correct. It appears as repeated because you made PDF file before stop trackiong the changes.
Point 3: Is this figure repeated?
Response 3: No it is correct. It appears as repeated because you made PDF file before stop trackiong the changes.
Point 4: …What is the relation between magnetic susceptibility, lithotypes and Au-grade?
Response 4: We explained it more in the Results and Discution section:
Borehole datasets show that the hydrothermal alteration has destroyed magnetite and replaced it with pyrite in the felsic volcanic rocks that are the host rocks of epithermal Au/Ag mineralization [27-30]. Therefore, the high-Au grades are accompanied by low-magnetic anomalies of the felsic rocks. This correlation made it possible to separate the high magnetics of the mafic volcanic rocks and dioritic intrusions from low-magnetic felsic volcanic rock and porphyritic intrusions based on the 3D cooperatively recovered magnetic susceptibility model. Consequently, we assigned three lithotype codes to quantify the lithological variations: Code 1 for felsic volcanic rocks, code 2 for mafic volcanic rocks, and code 3 for dioritic intrusions. The lithotype codes are matched and sorted according to their magnetic properties, from low magnetics of felsic volcanic rocks and intrusions to high magnetics of mafic volcanic rocks and diorites. The datasets are interpolated with the nearest neighbor (direct gridding) and kriging methods for comparison. Figure 5 shows that traditional in-terpolation methods cannot add any new geological information in places without bore-hole data.
Point 5: You give the conclusion before showing the results.
Response 5: Corrected.
Point 6: From this remark, we can deduced that your method is supervised learning. It would be appropriate to mention it at the beginning.
Response 6: Mentioned in the abstract and introduction.
Point 6: From this discussion, it seems that your two objectives you want to achieve with your study are lithotype reconstruction and Au-grade prediction. If it is the case, it should be clearly stated at appropriate places in your paper so that the reader does not get lost.
Response 6: Corrected.
